



# Analysis of secondary inorganic aerosols over the Greater Area of Athens using the EPISODE-CityChem source dispersion and photochemistry model

Stelios Myriokefalitakis[1], Matthias Karl[2], Kim A. Weiss[3,1], Dimitris Karagiannis[1], Eleni Athanasopoulou[1], Anastasia Kakouri[1], Aikaterini Bougiatioti[1], Eleni Liakakou[1], Iasonas Stavroulas[1], Georgios Papangelis[1], Georgios Grivas[1], Despina Paraskevopoulou[1], Orestis Speyer[1], Nikolaos Mihalopoulos[1], and Evangelos Gerasopoulos[1]

[1] Institute for Environmental Research and Sustainable Development (IERSD), National Observatory of Athens, Penteli, Greece
[2] Helmholtz-Zentrum Hereon GmbH, Geesthacht, Germany
[3] Natural Sciences, Institute of Physics and Meteorology, Earth and Climate System Sciences, Universität Hohenheim, Stuttgart, Germany

*Correspondence to*: Stelios Myriokefalitakis (steliosm@noa.gr) and Matthias Karl (matthias.karl@hereon.de)

**Abstract.** Secondary inorganic aerosols (SIA) are major components of fine particulate matter ($PM_{2.5}$), having substantial implications for climate and air quality in an urban environment. In this study, a state-of-the-art thermodynamic model has been coupled to the source dispersion and photochemistry city-scale chemistry transport model EPISODE-CityChem, able to simulate pollutants on a horizontal resolution of 100 x 100 $m^2$, to determine the equilibrium between the inorganic gas and aerosol phases over the Greater Area of Athens, Greece, for the year 2019. In agreement with in-situ observations, sulfate ($SO_4^{2-}$) is calculated to have the highest annual mean surface concentration (2.15 ± 0.88 µg m$^{-3}$) among SIA in the model domain, followed by ammonium ($NH_4^+$; 0.58 ± 0.14 µg m$^{-3}$) and fine nitrate ($NO_3^-$; 0.24 ± 0.22 µg m$^{-3}$). Simulations denote that $NO_3^-$ formation strongly depends on the local nitrogen oxide emissions, along with the ambient temperature, the relative humidity, and the photochemical activity. Additionally, we show that anthropogenic combustion sources may have an important impact on the $NO_3^-$ formation in an urban area. During the cold period, the combined effect of decreased temperature in the presence of non-sea salt potassium favors the partitioning of $HNO_3$ in the aerosol phase in the model, raising the $NO_3^-$ formation in the area. Overall, this work highlights the significance of atmospheric composition and the local meteorological conditions for the equilibrium distribution of nitrogen-containing semivolatile compounds and the acidity of inorganic aerosols, especially in urban areas where atmospheric trace elements from natural and anthropogenic sources coexist.



# 1 Introduction

Fine particulate matter (PM2.5) impacts the planet's radiative balance (Myhre et al., 2014) and severely affects air quality (WHO, 2021). Such effects strongly depend on the chemical and physical properties of PM2.5, e.g., shape, size distribution, and hygroscopicity. Among other particulate pollutants, like organic matter, elemental carbon, dust particles, and sea spray, secondary inorganic aerosols (SIAs) are identified as a critical PM2.5 fraction in an urban atmosphere. SIAs are formed in the atmosphere via gas-to-particle conversion of precursors of anthropogenic (e.g., coal combustion, biomass burning, vehicular),

biomass burning, and natural (e.g., marine, soil, lightning, volcanoes) origin, with sulfate ($SO_4^{2-}$), nitrate ($NO_3^-$), and ammonium ($NH_4^+$) being the three major SIA species. However, primary emissions, such as those from residential combustion, have been found to increase their atmospheric abundance (Zhang et al., 2023). SIAs may further contribute significantly to air quality degradation; for example, sulfates can reduce visibility and contribute to acid rain (Liu et al., 2019), affecting forest and aquatic ecosystems (Driscoll et al., 2003, 2016), and damaging building materials (Ozga et al., 2013). On the other hand,

SIAs are also linked to increased asthma exacerbation (Sarnat et al., 2015) and cardiovascular hospitalization (Basagaña et al., 2015) and can contribute to human mortality via the enhancement of fine particulate air pollution (Pope et al., 2020; Franklin et al., 2008). Focusing, in particular, on polluted environments, Nenes et al. (2020) indicated that atmospheric acidity, aerosol liquid water content (LWC), and meteorological conditions (temperature and relative humidity) are the main parameters that control SIA formation sensitivity, depending on ammonia ($NH_3$) and nitric acid ($HNO_3$) availability.

Sulfates are primarily formed by the oxidation of sulfur dioxide ($SO_2$) as a byproduct of fossil fuel burning, such as coal and oil in power plants and industrial processes, as well as residual oil burning in shipping. In the gas phase, the $SO_2$ homogeneous oxidation via OH radicals during daytime strongly depends on temperature, while the heterogeneous oxidation of $SO_2$ via $H_2O_2/O_3$ may proceed in aqueous media such as cloud liquid water and in-cloud scavenging processes (Seinfeld and Pandis, 2006), enhanced in the presence of transition metal ions (e.g., $Mn^{2+}$ and $Fe^{3+}$; Alexander et al., 2009), and/or on aerosol surfaces

such as dust aerosols (Wang et al., 2022). Aqueous phase processes are nevertheless found to be responsible for about two-thirds of the total sulfate production in the global troposphere (e.g., Myriokefalitakis et al., 2022). Overall, it is well established that the amounts of precursor gases and oxidants, the cloud cover, the amount of available surface area, and meteorological factors such as temperature and relative humidity all affect the rate of $SO_2$ conversion to sulfate (Liu et al., 2021).

Particulate nitrate formation depends on the emissions of nitrogen oxides ($NO_X$) from traffic, power plants, and industrial
activities, along with the gas-phase concentrations of ammonia ($NH_3$) as emitted by agricultural activities, waste management, industrial processes, and the biosphere. The formation of $NO_3^-$ is, however, a rather complex process, strongly dependent on interactions of acidity, relative humidity, and temperature (Guo et al., 2017b; Nenes et al., 2020), with the partitioning of $HNO_3$ to the particulate phase enhanced under lower ambient temperatures (Guo et al., 2017b). Nevertheless, since ammonium nitrate ($NH_4NO_3$) is formed when $NH_3$ is in excess upon the formation of ammonium sulfate and ammonium bisulfate, such an

equilibrium, besides the availability of $NH_3$, also depends on the concentration of sulfuric acid ($H_2SO_4$). The relative humidity is, nevertheless, a key factor for SIA formation, determining the aerosol LWC and thus the aerosol acidity levels, which in



turn impact the HNO$_3$ partitioning in the aerosol phase. All in all, NO$_3^-$ formation increases the aerosol LWC by lowering aerosol acidity via dilution and thus further promoting HNO$_3$ partitioning into the particulate phase (Guo et al., 2017a).

Thermodynamic calculations (Fountoukis and Nenes, 2007; Nenes et al., 1998; Clegg et al., 1998) are considered to be the appropriate way of assessing non-linearities in an inorganic aerosol system and the behavior of a non-ideal liquid solution. Such methods can explicitly account for the equilibrium state of inorganic particles by calculating the gas- and particulate-phase fractions of the inorganics, along with the corresponding aerosol LWC and the respective levels of acidity (pH), both of which are very important for assessing the burden of the secondary inorganic PM fraction. The presence of nonvolatile cations (NVCs) in thermodynamic calculations, however, has been shown to substantially affect the ion balance (Guo et al., 2018) and thus the partitioning of HNO$_3$/NO$_3^-$ and NH$_3$/NH$_4^+$ species, especially in areas with abundant mineral dust and sea spray aerosols (Athanasopoulou et al., 2008, 2016), such as the Eastern Mediterranean. For example, calcium-containing dust particles may react with nitric acid to form calcium nitrate (Ca(NO$_3$)$_2$), significantly enhancing NO$_3^-$ formation, especially when natural and anthropogenic particulate matter coexist. On the other hand, NVCs from combustion sources related to natural or anthropogenic activities may also critically impact the predictions of inorganic partitioning between the gas and the particulate phase in an urban environment (Cheng et al., 2022).

Regional chemistry transport models (CTM) are promising tools for providing spatial information on environmental impacts occurring in the urban landscape. Although such modeling tools can successfully simulate the inorganic and organic components of atmospheric aerosols at the background scale (e.g., Brandt et al., 2003; Athanasopoulou et al., 2013; Zakoura and Pandis, 2018; Basla et al., 2022), they are not rather accurate in the representation of their spatial distribution in proximity to point and/or line sources (Santiago et al., 2022). On the contrary, urban-scale CTM models have the advantage of mapping air pollution levels in high-resolution ($\leq$ 1km) (e.g., Hurley et al., 2005; Karl et al., 2019; Denby et al., 2020), although critical processes related to air quality, such as aerosol representation and its chemical speciation, may still be simplistically represented due to the high computational load of such modeling systems. Indeed, urban-scale CTMs usually estimate the bulk mass of particulate matter (PM$_{2.5}$ and/or PM$_{10}$) without having incorporated their chemical interactions with gaseous pollutants in the atmosphere, like in the case of inorganic chemical components in the aqueous and/or solid state. Such a simplistic aerosol parameterization in local/city-scale models, which still remains a common approach, nevertheless represents the next challenging issue of multi-scale modeling (Sokhi et al., 2022).

Several regional-scale numerical simulations of atmospheric pollution have been performed over the Eastern Mediterranean area (Fountoukis et al., 2011; Im et al., 2013; Athanasopoulou et al., 2015; Kushta et al., 2019; Kontos et al., 2021; Liora et al., 2022), focusing in particular on the urban area of Athens, one of the most polluted cities in the EU. Most of these studies highlight the complex interactions among nitrogen and sulfur oxides of anthropogenic origin with sea salt and/or dust particles of natural origin over the area (Athanasopoulou et al., 2008; Karydis et al., 2016). These findings are also confirmed by several local observational studies (Bardouki et al., 2003; Koçak et al., 2004, 2012; Stavroulas et al., 2021; Dimitriou et al., 2021). On the other hand, recent city-scale modeling studies over Athens (Grivas et al., 2020; Ramacher et al., 2021; Gratsea et al., 2021;





Lasne et al., 2023) clearly demonstrate the difficulties in simulating the inorganic constituents of atmospheric aerosols in the area, in particular at the intra-urban scale.

To tackle the above challenges in the context of multi-pollutant studies at a city-scale spatial resolution, we here develop a framework for fine aerosol thermodynamic calculations to better understand the SIA formation mechanisms in an urban environment. Section 2 provides an overview of the model, focusing mostly on the new developments applied in this study. In

particular, we describe the aqueous-phase chemistry scheme used to simulate the atmospheric $SO_4^{2-}$ production in cloud droplets, the respective developments for the thermodynamic calculations of semi-volatile inorganics, along with a series of code updates currently applied in the model. In Sect. 3, we present the model-derived $SO_4^{2-}$, $NO_3^-$, and $NH_4^+$ surface concentrations and their evaluation against in situ observations. Finally, in Sect. 4, we discuss the simulated processes and summarize the implications of explicitly resolving the secondary formation of $PM_{2.5}$ in a city-scale chemistry transport model

for air quality, as well as the plans for future model development.

## 2. Methodology and Data

The numerical atmospheric model system used for this study is the chemistry-transport model (CTM) EPISODE-CityChem (Karl et al., 2019), coupled here to atmospheric processes relevant to SIA formation. The model is applied over the Greater Area of Athens (Greece), centered at the urban center (Fig. 1), horizontally covering a domain of 45 x 45 $km^2$ (SW corner

23.4E°, 37.8N°, 1 x 1 $km^2$ cell size, with a subgrid of 100 m) and vertically representing a 24-layered atmosphere up to 3.7 km. For the current study, simulations of air quality are performed for 2019. Three simulations have been conducted in this work to investigate uncertainties regarding the impact of SIA formation in the Greater Area of Athens (GAA): i) the base simulation where a complete chemistry/thermodynamic scheme is accounted in the model (W/ SIA); ii) a simulation where the secondary production of $SO_4^{2-}$, $NO_3^-$, and $NH_4^+$ is neglected (W/O SIA), that is focused on the impact of long-range

transport to SIA concentrations in the study area; and iii) an extra simulation where the impact of NVC emissions related to domestic burning on the SIA formation is neglected (W/O Kbb), focusing thus on the importance of potassium ($K^+$) on the equilibrium distribution of nitrogen-containing semivolatile compounds. Finally, available in situ measurements for the area and year of study are used to assess the accuracy of model predictions and the efficiency of the performed advancements.

### 2.1 The EPISODE-CityChem model

EPISODE-CityChem is a city-scale CTM designed for treating complex atmospheric chemistry in urban areas and improving the representation of near-field dispersion. EPISODE is an Eulerian dispersion model developed by the Norwegian Institute for Air Research (Slørdal et al., 2007) to simulate pollutant dispersion at the city scale and microscale simultaneously. EPISODE consists of a 3-D Eulerian grid that interacts with a subgrid Gaussian dispersion model for the dispersion of pollutants emitted from both line and point sources (Hamer et al., 2020), allowing the retrieval of pollutant concentrations at

the subgrid scale in an urban area. The CityChem extension offers an explicit description of the gaseous reactive pollutants'



photochemistry on the Eulerian grid, along with the dispersion close to point emission sources (e.g., industrial stacks) and line emission sources (open roads and streets). For this work, however, supplementary emissions are incorporated into the model domain (see Sect. 2.1.3) to enable a realistic simulation of SIA formation.

### 2.1.1 Gas-phase chemistry

EPISODE-CityChem has several schemes for solving the chemistry of the gas phase. For the standard model configuration, the updated chemistry scheme, EmChem09-HET, is applied on the coarse grid, which supports more than 100 reactions, including photodissociation, thermal, and heterogeneous reactions, and about 70 species. However, for this study, the well-documented Carbon Bond Mechanism 2005 (CB05; Yarwood et al., 2005), along with more recent modifications (mCB05) as introduced by Williams et al. (2013, 2017) are applied. For comparison, we note that the mCB05 configuration here uses 50 tracers with roughly 134 reactions, respectively. The Kinetic PreProcessor (KPP) software version 2.2.3 (Damian et al., 2002; Sandu and Sander, 2006) is employed to integrate the mCB05 gas-phase chemical mechanism for this work, based on the generated Rosenbrock solver. A more detailed description and evaluation of the mCB05-KPP implementation can be found in Myriokefalitakis et al. (2020).

Photodissociation rates were calculated inline using the Tropospheric Ultraviolet and Visible (TUV) Radiation model (Lee-Taylor and Madronich, 2002). The TUV module calculates actinic flux and photolysis rates in each vertical layer within a 1-D column of the Eulerian grid. The implemented wavelength grid ranges from 170-735 nm with 138 wavelength bins; 31 photodissociation reactions (out of 113 available in TUV) were selected for the implemented TUV module, covering all relevant photolysis rates in the lower troposphere. Additionally, the model includes a light-dependent ground surface source of nitrous acid (HONO) on top of the respective direct emissions from vehicles (Karl and Ramacher, 2021). In more detail, the ground surface reaction producing HONO was parameterized according to Zhang et al. (2016) with an NO uptake coefficient (gamma) depending on the light intensity, i.e., when light intensity is less than 400 W m$^{-2}$ a gamma value of $2 \times 10^{-5}$ is used; otherwise, a factor of light intensity/400 is used to scale the gamma value.

Pollutant concentrations in the sub-grid components (i.e., the Gaussian models for line and point source dispersion) are determined by the EP10-Plume scheme, which contains 10 compounds and 17 reactions of $O_3$, NO, $NO_2$, $HNO_3$, CO, and formaldehyde (HCHO). In the line source sub-grid model, the simplified street canyon model (SSCM) is applied to calculate pollutant dispersion in street canyons, and in the point source sub-grid model, the WMPP (WORM Meteorological Pre-Processor) is integrated to calculate the wind speed at plume height. The rate of change of the concentrations of NO, $NO_2$, and $O_3$ in a street canyon is described in terms of chemical reaction, direct emissions, and background concentration. The mixing processes are parameterized via the mixing time, which corresponds to the typical residence time of pollutants in a street canyon. More details on the sub-grid photochemistry schemes of EPISODE-CityChem can be found in (Karl et al., 2019).



### 2.1.2 Extensions to the aerosol scheme

EPISODE-CityChem is extended for this work with online calculations of the main inorganic aerosol components, $SO_4^{2-}$, $NO_3^-$, and $NH_4^+$, in the presence of non-volatile cations from sea salt, dust, and domestic heating. For this, the aqueous-phase formation of $SO_4^{2-}$ in cloud droplets and the thermodynamic calculations, which determine the gas/particle partitioning of $NH_3/NH_4^+$ and $HNO_3/NO_3^-$, are explicitly considered in the model. Aqueous phase oxidation of the dissolved $SO_2$ via $H_2O_2$ and $O_3$ in cloud droplets is added to the model. $H_2O_2$ produced in the gas phase can be rapidly dissolved in the liquid phase due to its high solubility, and the dissolved $H_2O_2$ reacts rapidly with the $HSO_3^-$. Note that the dissociation of $H_2O_2$ is neglected here. $SO_2$ oxidation in the aqueous phase is much faster than its gas phase homogeneous oxidation via OH radicals, with the dissolved $H_2O_2$ being the most effective oxidant in the aqueous phase (Seinfeld and Pandis, 2006), especially when the solution becomes acidic. At a higher pH (pH > 4), the dissolved $SO_2$ oxidation by ozone ($O_3$) tends to dominate the S(IV) oxidation, even under dark conditions. Although laboratory studies have shown that dissolved $SO_2$ can also be oxidized in the presence of transition metals via other pathways (Harris et al., 2013), such reactions are not currently considered in our model. Lastly, the acidity of clouds is calculated using the electro-neutrality of strong acids (i.e., $H_2SO_4$, $SO_4^{2-}$, $HNO_3$, $NO_3^-$) and $NH_4^+$, along with the dissociations of $CO_2$, $SO_2$, and $NH_3$. The reaction rates and the constants used to calculate the sulfur chemistry in the model are presented in Table 1.

The ISORROPIA II thermodynamic equilibrium model (Fountoukis and Nenes, 2007) has been coupled to EPISODE-CityChem to calculate the gas/particle partitioning of inorganic aerosols. ISORROPIA-II calculates the gas–liquid–solid equilibrium partitioning of the $K^+$ - $Ca^{2+}$ - $Mg^{2+}$ - $NH_4^+$ - $Na^+$ - $SO_4^{2-}$ - $NO_3^-$ - $Cl^-$ - $H_2O$ aerosol system, and it is applied here in the forward mode. NVCs from dust and sea salt aerosols are taken into account in the thermodynamic calculations considered here as a fraction of $PM_{2.5}$ concentration. It is here assumed that constant mass ratios correspond to dust concentrations for $Ca^{2+}$, $Na^+$, $K^+$, and $Mg^{2+}$ of 1.2%, 1.5%, and 0.9%, respectively (Sposito, 1989). Accordingly, for sea spray aerosols, mean mass fractions of 55.0% $Cl^-$, 30.6% $Na^+$, 7.7% $SO_4^{2-}$, 3.7% $Mg^{2+}$, 1.2% $Ca^{2+}$, and 1.1% $K^+$ are applied (Seinfeld and Pandis, 2006).

Inorganic particles can be solid and/or composed of an aqueous supersaturated solution. The assumption of aqueous aerosol for the acidity-sensitive semivolatile inorganic species (e.g., $NH_3/NH_4^+$ and $HNO_3/NO_3^-$) is found, though, to perform quite well with a series of aerosol LWC observations when the relative humidity is above 40% (Bougiatioti et al., 2016; Guo et al., 2015, 2017a). However, Cheng et al. (2022) recently indicated that although under high relative humidity conditions, both assumptions may perform almost identically, the aqueous aerosol setup might perform better under high temperatures (>10 °C) and low relative humidity (<60%) conditions, while the solid aerosol setup might perform better for lower temperatures. For this, the metastable aerosol setup is here applied for temperatures higher than 10 °C and RH lower than 60% or higher than 83%. On the other hand, the stable aerosol setup is applied mostly at lower temperatures (<5 °C). We note, however, that in our case, there were only minimal differences (less than 1% in the simulated SIA concentrations during the cold period) between simulations with only metastable (liquid) and variable aerosol state assumptions in thermodynamic calculations.



Even though this work is focused on the SIA fine aerosol fraction, nitrate aerosols are explicitly calculated for both the fine
and coarse modes (i.e., unlike $SO_4^{2-}$ and $NH_4^+$). The rationale behind this is that nitrate is found to significantly contribute to
the coarse aerosol sizes (Koulouri et al., 2008), indicating that a bulk approach may lead to an overestimation of the $HNO_3/NO_3^-$
partitioning in the model, especially in areas with intense NVC contribution, such as from sea salt and dust aerosols, and from
anthropogenic combustion. For this practical application, kinetic limitations (Pringle et al., 2010) by mass transfer and transport
between the nitric acid in the gas phase and the particulate nitrate in $PM_{2.5}$ and $PM_{10}$ are here considered, with ISORROPIA-
II then re-distributing the respective masses between the gas and the aerosol phases. We note, however, that the coarse nitrate
is not further taken into account in the thermodynamic calculations, assuming for simplicity that the gaseous $HNO_3$, able to
condense in the coarse aerosol mode, partitions directly into the particulate bulk phase.

### 2.1.3 Emissions

Anthropogenic emissions were originally provided by the Copernicus Atmosphere Monitoring Service (CAMS) and then
improved over the domain of interest with the spatial disaggregation approach and tool documented in (Ramacher et al., 2021).
CAMS provides the annual emission rates of CO, $NH_3$, NMVOCs, $NO_X$, $PM_{10}$, $PM_{2.5}$, and $SO_2$ from the road, air, rail transport,
navigation, mobile machinery, fuel production, industrial (paper, cement, minerals, metals, etc.), stationary combustion,
agriculture, waste, solvent use, and public power sectors. The dataset adopts the GNFR sector classification and provides
gridded data at 0.1 x 0.05 degrees. This dataset is then combined with GNFR-dependent high-resolution spatial proxies (based
on data from E-PRTR, GHS-POP, CLC, and OSM) towards the spatial disaggregation in 1x1 $km^2$ (area sources), point
(industrial), and line emissions (road and railway emissions).

Sea salt emissions are calculated for this work following the parameterizations from Gong (2003) and Vignati et al. (2010).
The flux of the number of sea-spray particles is expressed as a function of the particle radius at 80% humidity and the 10 m
horizontal wind speed, assuming number median dry radius values of 0.09 and 0.794 µm for fine and coarse sea-spray particles,
along with geometric standard deviations of 1.5 and 2.0, respectively (Vignati et al., 2010). To further address the temperature
effects on the sea spray source fluxes, polynomial expressions derived based on laboratory (chamber) experiments (Salter et
al., 2015) have been implemented. Note here that the emitted sea-spray particles are assumed to consist of pure sodium chloride
(NaCl), although an explicit sea-salt composition is applied for thermodynamic calculations in the model (see Sect. 2.2.2).

Potassium ($K^+$) emissions from anthropogenic wood-burning processes are further considered in our calculations, owing to
their significance for the thermodynamic equilibrium distribution of nitrogen-containing semivolatile compounds. For this, we
apply monthly factors to the $PM_{2.5}$ emissions from the domestic heating sector to derive the respective particle $K^+$ fluxes. The
non-sea salt $K^+/PM_{2.5}$ concentration ratios are derived from filter-based $PM_{2.5}$ measurements in the center of Athens
(Paraskevopoulou et al., 2014), specifically at the Thissio supersite (see Sect. 2.3). Assuming that in domestic wood
combustion episodes $K^+$ is directly emitted as part of the $PM_{2.5}$, an average fraction of 3.52% w/w contribution is here applied
to the respective $PM_{2.5}$ emissions for the winter months, as obtained by a positive matrix factorization (PMF) in central Athens
(Thissio) from December 2013 to March 2016 (Theodosi et al., 2018). $SO_4^{2-}$ primary sources from residential combustion are





also accounted for in the model, owing to the significance of their ambient concentrations, accounting here for a fraction of 3.7% w/w (Zhang et al., 2023) of the $PM_{2.5}$ emitted during domestic wood combustion episodes. We note, however, that although Zhang et al. (2023) indicate that $NO_3^-$, and $NH_4^+$ can also be emitted from domestic combustion processes, the

respective emission factors are found to be 3–4 times lower compared to $SO_4^{2-}$, i.e., accounting for only up to 1% of the $PM_{2.5}$ emitted mass. For this reason, no primary emissions for $NO_3^-$, and $NH_4^+$ are accounted for in this study.

### 2.1.4 Boundary conditions

The CAMS European regional ensemble reanalysis data is used to create boundary conditions for $O_3$, NO, $NO_2$, CO, $NH_3$, $PM_{2.5}$, $PM_{10}$, PAN, $SO_2$, and NMVOC concentrations (chemically split to the species treated by the local model). Due to the

chemical advancements of the current work (SIA treatment), the CAMS global atmospheric composition forecast supports the boundary conditions for $H_2O_2$, $N_2O_5$, $HNO_3$, methacrolein/methyl vinyl ketone, $N_2O_5$, $HNO_3$, and $HNO_4$, along with sea salt aerosols in the fine and coarse sizes (not available on the regional scale). Furthermore, since the CAMS European regional ensemble reanalysis data only provide SIA concentrations as lumped species, we also use the respective global atmospheric composition forecast to derive the individual species (i.e., $SO_4^{2-}$, $NO_3^-$, and $NH_4^+$) normalized to the regional ensemble models'

SIA concentration data. The same assumption is applied for dust aerosols since dust concentrations are only provided as a part of the $PM_{10}$ fraction in the regional reanalysis data. $NO_3^-$ in the coarse mode is directly derived from the global forecast data (where available). The boundary conditions interface (BCONCAMS) developed at Helmholtz-Zentrum Hereon was utilized for the interpolation of the CAMS regional ensemble and global forecast product and the mapping of the chemical species to the urban air quality model. The interface performs bilinear interpolation in the horizontal and linear interpolation in the vertical

and projects the pollutant concentrations to the city domain. We note that several utilities allow the preparation of input files or the conversion of these to the necessary CityChem input formats. More details on the utilities used to produce the EPISODE-CityChem input files can be found in Karl et al. (2019).

### 2.1.5 Meteorology

The CTM EPISODE-CityChem is offline coupled with the mesoscale numerical weather prediction system Weather Research

and Forecasting (WRF). The meteorological simulations are initialized by the synoptic-scale meteorological reanalysis ensemble means (ECMWF ERA5). WRF is applied with a telescoping nesting (3 domains), with the innermost domain covering the CTM area with a 1x1 $km^2$ spatial resolution. Hourly varying meteorological variables used as inputs in the EPISODE-CityChem include pressure, air temperature, temperature gradient, relative humidity, sensible and latent heat fluxes, total solar radiation, cloud fraction, and the cloud LWC.

### 2.2 In-situ measurements

In-situ aerosol chemical composition measurements conducted at 1) the Thissio Air Monitoring Supersite and 2) the Piraeus monitoring station, both operated by the National Observatory of Athens, are here used to evaluate the model's ability to



simulate the main SIAs ($SO_4^{2-}$, $NH_4^+$, and $NO_3^-$) in the GAA, Greece. The Thissio Air Monitoring Supersite (37.9732° N, 23.7180° E, 105 m a.s.l.) is located on the premises of the National Observatory of Athens, in central Athens. The site is far

from major traffic roads, in a moderately populated residential area, and thus is considered representative of urban background conditions in the center of the Athens basin (Grivas et al., 2019; Panopoulou et al., 2020), as it serves as a receptor of pollutants from different sources and with various degrees of processing. Online measurements took place throughout 2019 at Thissio, apart from the period when the instrumentation for online aerosol SIA measurements was moved to the Piraeus site. Measurements were conducted at a central location during two periods in winter (10 December 2018 - 16 January 2019) and

summer (11 June 2019 - 9 July 2019). The instruments in Piraeus were located on the 1st floor (9m above ground) of the Athens Metro terminal station building (Urban Rail Transport S.A., Athens, Greece; 37.9479º N, 23.6429º E, 10 m a.s.l.). The site is influenced to an extent by traffic emissions, being at a distance of 70 m from the coastal avenue in the Piraeus port area (Stavroulas et al., 2021).

High-temporal resolution measurements of non-refractory submicron aerosol chemical composition at both sites were

performed with a quadrupole Aerosol Chemical Speciation Monitor (ACSM; Aerodyne Research Inc.; Ng et al., 2011), operated at a 30-minute temporal resolution. The instrument provides concentrations of ammonium, sulfate, nitrate, chloride, and organic matter. Details on the limits of detection, sampling, and calibration of the instrument can be found in Stavroulas et al. (2019, 2021). QA/QC of ACSM measurements included the comparison of the daily-averaged online and offline measurements. The estimation of sea salt, non-sea salt potassium, and dust was performed using the methodology of Sciare et

al. (2005). Although the evaluation of dust concentrations is out of the scope of this study, we note that the estimation of atmospheric dust levels using the calcium ion as a proxy (Sciare et al., 2005) may be largely uncertain due to major contributions from other sources. It should also be noted that since ion chromatography quantifies mostly the inorganic components, it is expected that interferences may occur for sulfate and nitrate, especially during winter, when nitrites and organic nitrates and sulfates are variably present (Guo et al., 2016).

## 3 Results and discussion

### 3.1 Model evaluation

#### 3.1.1 Sulfate

Observations indicate that the hourly $SO_4^{2-}$ concentrations present only a slightly increasing trend from winter to summer 2019 over Athens (Fig. 2a). During the warm season, however, the scarcity or absence of precipitation in the region, along with

increased transport through the boundaries and mostly the higher oxidation potential, resulted in an overall increase in $SO_4^{2-}$ concentrations in the model (W/ SIA), although still lower compared to observations. The enhanced summer concentrations in the model are nonetheless in accordance with previous observational and modeling studies in the area (Paraskevopoulou et al., 2015; Grivas et al., 2018; Theodosi et al., 2018; Athanasopoulou et al., 2015), suggesting that the majority of $SO_4^{2-}$ mainly



originates from long-range transport (e.g., Balkan countries and the wider Black Sea region) due to fossil fuel combustion in power production and other industrial plants (Aksoyoglu et al., 2017). This is also clearly shown by the present sensitivity model calculations, indicating that when the local $SO_4^{2-}$ production via aqueous-phase $SO_2$ oxidation is neglected (W/O SIA), the simulated $SO_4^{2-}$ concentrations remain practically unchanged in most of the cases, especially during the warm periods. EPISODE-CityChem captures the observed hourly $SO_4^{2-}$ surface levels in Piraeus and Thissio, with a correlation coefficient (r) of 0.46 and an annual mean bias of -1.2 μg m$^{-3}$ (Fig. 2a). The model tends to overall underestimate the observations, presenting for 2019 a negative normalized mean bias (nMB) of -42%. Our simulations nevertheless indicate that the underestimation of the modeled $SO_4^{2-}$ concentrations should be mainly due to the too low long-range transport in the area through the boundaries, since only a weak local $SO_4^{2-}$ formation is supported by the model. Indeed, when comparing the base simulation (W/ SIA) with the W/O SIA sensitivity simulation, where the local $SO_4^{2-}$ production in cloud droplets is neglected, only a slight increase in the surface $SO_4^{2-}$ concentrations is illustrated, without any noteworthy change (< 1%) in the respective correlation statistics. On the other hand, the underrepresentation of $SO_x$ (= $SO_2$ + $SO_4^{2-}$) local emissions, or even a too slow $SO_2$ oxidation in the gas phase, could further contribute to the model's underestimation of the observed values.

### 3.1.2 Nitrate

The observed and modeled $NO_3^-$ concentrations increase during the cold period (Fig. 2b), presenting a pronounced seasonality. The increase mentioned above is caused by both the formation of $NH_4NO_3$ under low ambient temperatures (Theodosi et al., 2018) and the presence of non-volatile potassium cations from domestic heating processes, which can significantly promote the formation of $KNO_3$. On the contrary, the observed $NO_3^-$ concentrations are considerably reduced in summer (Paraskevopoulou et al., 2015; Liakakou et al., 2022), owing to the thermal instability and volatilization of the $NH_4NO_3$. Sensitivity model simulations clearly indicate that $NO_3^-$ is a local pollutant rather than a transported one in the GAA. This agrees with previous modeling studies where road traffic is found to significantly contribute to $NO_3^-$ in eastern Europe (Aksoyoglu et al., 2017). Indeed, when the thermodynamics are neglected in model calculations (W/O SIA), $NO_3^-$ concentrations are severely underestimated over GAA in almost all seasons of the year, except for some cases during the summer months (July and, to a lesser extent, August). Although EPISODE-CityChem tends to calculate very low $NO_3^-$ levels during the warm period for the base simulation (W/ SIA), the maxima during July and August due to the contribution of long-range transport from biomass burning processes (such as wildfires in Attica and the island of Evia in July and August 2019) seem not to be well captured by the model. On the other hand, the W/O Kbb sensitivity model simulation clearly indicates that the presence of $K^+$ associated with domestic heating processes plays a crucial role in $NO_3^-$ formation during the winter period, leading overall to a lower nMB compared to the base simulation (i.e., -63% vs. -61%). This, yet, is in accordance with previous findings in the region, where a good correlation between $K^+$ with OC, EC, and $NO_3^-$ has been observed during winter (Theodosi et al., 2018; Fourtziou et al., 2017), thus clearly supporting their common origin during the cold period. In more detail, residential biomass burning processes are found to be directly responsible for the winter maxima measured over GAA, while intense Saharan dust episodes under the local southerly winds may also have an impact on the spring peak.



A better correlation compared to that of $SO_4^{2-}$ is found when comparing the model and the measurements for the hourly $NO_3^-$ surface levels (r = 0.58). The model, however, tends to overall underestimate the observations, with a slightly negative bias of around -0.4 µg m$^{-3}$ and an nMB of roughly -61% for the year 2019, mainly due to the too low summertime simulated concentrations (Fig. 2b). We note, nevertheless, that an underestimation of the observed concentrations is here expected since, on the one hand, some ammonia sources, such as from agricultural soils, are currently omitted in the model, and on the other hand, species that may be accounted for in the observational data, such as nitrites and organic nitrates that may also be present during winter and partition in the particulate phase (Guo et al., 2016), are not yet included in the presented chemical mechanism. Still, correlation statistics clearly denote that the local $HNO_3$ partitioning in the particulate phase is the main pathway of fine particulate $NO_3^-$ formation in the study area, and more specifically, that the presence of $K^+$, either from domestic burning processes or dust events, plays a crucial role in the $NO_3^-$ concentration distributions.

### 3.1.3 Ammonium

The observed $NH_4^+$ concentrations (Fig. 2c) generally present a similar seasonal pattern to that of $SO_4^{2-}$, although somehow with a more intense trend during the warm season. As already found in previous works (Mantas et al., 2014; Paraskevopoulou et al., 2015; Theodosi et al., 2018), observations of $NH_4^+$ and $SO_4^{2-}$ are highly correlated (r = 0.79) indicating that nss-$SO_4^{2-}$ can be considered at least partially neutralized by $NH_4^+$. The simulated sulfate/ammonium ratio for the base simulation (r = 0.73) shows that sulfate exists mainly in the model in the form of $NH_4HSO_4$ and $(NH_4)_2SO_4$ in the two sites (i.e., Thissio and Piraeus), in agreement with previous studies in the wider eastern Mediterranean area (Fourtziou et al., 2017; Koulouri et al., 2008; Bardouki et al., 2003; Athanasopoulou et al., 2008). Nevertheless, during the cold season, $NH_4^+$ is also found to be significantly correlated with $NO_3^-$ (r = 0.62), indicative of $NH_4NO_3$ formation, respectively. Note, however, that for this work, the non-sea salt chloride is not considered in model calculations, so the nss-$NH_4Cl$ does not contribute to the calculated $NH_4$ concentrations.

The base case simulation of EPISODE-CityChem captures the observed $NH_4^+$ hourly surface observations in Piraeus and Thissio with a correlation coefficient (r) of 0.57 and a negative mean bias of -0.4 µg m$^{-3}$ (Fig. 2f), with a calculated negative nMB of -46% for 2019. However, when the thermodynamic calculations are neglected in the model in the W/O SIA simulation (i.e., accounting only for the transport from the model's boundaries contributing to the $NH_4^+$ levels), the agreement tends to improve, leading to a correlation coefficient (r) of 0.61, a negative mean bias of -0.2 µg m$^{-3}$, and a negative nMB of -22% (Fig. 2f). This can be explained due to the volatilization of $NH_4^+$ for the base simulation when thermodynamics are taken into account in the model, which overall shifts $NH_4^+$ to gas-phase $NH_3$ (Guo et al., 2018).

The significance of aerosol pH levels on $NH_4^+$ volatilization is further supported by the model sensitivity simulation when the contribution of $K^+$ from domestic heating processes is neglected in the thermodynamic calculations (W/O Kbb). Indeed, the absence of $K^+$ increases the acidity in the aerosol phase and thus enhances the gas/particle partitioning of $NH_3/NH_4^+$. Overall, the improved correlation of model results with observations in the absence of $K^+$ from domestic heating compared to the base case simulation (i.e., r = 0.61, bias = -0.3, and nMB = -37%) indicates that the model misses primary $NH_3$ emissions from



sources like road transport, that could contribute up to 10% to $NH_4^+$ ambient levels (Aksoyoglu et al., 2017), or from non-urban sources, such as the volatilization of animal waste and synthetic fertilizers, biomass combustion, and losses from natural soils (e.g., Fameli and Assimakopoulos, 2016)

### 3.2 Spatial model concentration fields

The mapping of the main SIAs at ground level, as simulated by EPISODE-CityChem averaged for the winter (January,
February, and December 2019) and summer (June, July, and August 2019) periods, is presented in Fig. 3. During both seasons, high $SO_4^{2-}$ levels are simulated near the major emission hotspots, such as shipping routes around the Ports of Piraeus and Eleusis and large combustion plants in the Thriassion Plain (Fig. 1), where the vast majority of the surface $SO_2$ emissions in the GAA occur. During summer (Fig. 3b), the predicted $SO_4^{2-}$ surface concentrations are close to $2.86 \pm 0.92$ µg m$^{-3}$ in the center of Athens and the suburbs, while during winter (Fig. 3a) the season mean simulated concentrations vary between 1.31
and 4.45 µg m$^{-3}$. The results indicate that long-range transport is an important contributor to $SO_4^{2-}$ surface concentrations in the area, which are maximized during the summer due to enhanced photochemical activity. On average, $SO_4^{2-}$ is calculated to have an annual mean surface concentration of $2.15 \pm 0.88$ µg m$^{-3}$ for the year 2019 in the model domain. Sensitivity model calculations confirmed this, showing that chemical production in the model domain can support up to 10% of the wintertime $SO_4^{2-}$ surface concentrations (Fig. 4a), mainly in higher altitude areas, such as Parnitha Mt. (Fig. 1), where cloudiness is
enhanced and the increased cloud LWC promotes the aqueous phase oxidation of ambient $SO_2$. Around the city center, however, the chemical formation contribution to the surface $SO_4^{2-}$ levels is calculated to be rather low (roughly 4%).

The average fine $NO_3^-$ concentrations in the model are calculated to vary between 0.05 and 1.50 µg m$^{-3}$ during the cold period (Fig. 3c). A mean $NO_3^-$ concentration of $0.51 \pm 0.20$ µg m$^{-3}$ is simulated during the winter months, while only $0.05 \pm 0.05$ µg m$^{-3}$ is calculated during the summer (Fig. 3d). In the model domain, $NO_3^-$ registered an annual mean surface concentration of
$0.24 \pm 0.23$ µg m$^{-3}$ for 2019. In contrast, however, to the $SO_4^{2-}$ surface concentrations, $NO_3^-$ levels present notable spatial variability, with much higher concentrations calculated near rather than away from the city center during the winter, compared to the negligible concentrations in the warm season. This indicates that $NO_3^-$ formation significantly depends on the local meteorological conditions (i.e., temperature and relative humidity) along with the anthropogenic emissions of precursor species that are widespread in the GAA (Fig. 1), mainly from traffic, and especially the presence of NVCs such as K+ from domestic
heating in winter that can support an increase in $NO_3^-$ surface concentrations of about $10 \pm 4\%$ in the model domain (Fig. 4d). It is noted that in this study, $NO_3^-$ is only either formed in the model domain as a secondary product or transported from the boundaries, without any primary contribution. However, the enhanced concentrations around the city center in the model clearly indicate that $NO_3^-$ is mainly a local pollutant rather than a transported one over the GAA. The latter is further supported through sensitivity model calculations performed for this work, demonstrating that when thermodynamics are accounted for
in the model (Fig. 4c), more than 80% of the $NO_3^-$ surface concentrations on an annual basis in downtown Athens (Fig. 1) are directly attributed to the $HNO_3$ gas/particle partitioning.



The average surface $NH_4^+$ concentrations in the model are calculated to be without strong annual variability in the GAA for 2019. For the cold period (Fig. 3e), the model simulates that $NH_4^+$ concentration varies between 0.30 in the suburbs and 1.40 µg m$^{-3}$ around the city center, while for the summertime it is between 0.33 and 1.18 µg m$^{-3}$, respectively (Fig. 3f). For the year 2019, $NH_4^+$ is calculated to have an annual mean surface concentration of $0.58 \pm 0.14$ µg m$^{-3}$ in the model domain. Model simulations show that the available anthropogenic $NH_3$ emissions in the model domain tend to quickly neutralize the primary strong acids present in the urban atmosphere of Athens, like $H_2SO_4$ and $HNO_3$, which subsequently partition into the aerosol aqueous phase in the form of the ammonium ion. As already discussed in Sect. 3.1.3, when thermodynamics are included in model simulation, an increase in volatilization of $NH_4^+$ is calculated that, on an annual basis, can support a decrease in its concentrations up to 70% in the suburbs (Fig. 4e). On the other hand, much lower $NH_4^+$ volatilization is calculated around the Thriassion Plain, where combustion plants exist, as well as in the region's major port zones (Fig. 1), due to primary $NH_3$ sources. The latter indicates that although traffic, solid and water waste management facilities, and certain industrial processes (e.g., refineries, cement manufacturing, livestock production) seem to be the main $NH_3$ sources in urban areas (Liakakou et al., 2023), ammonia may also be strongly related to agricultural activities and soil sources in rural areas, which in turn significantly impacts $NH_4^+$ concentrations; however, $NH_3$ emissions from such sectors are not currently included in our model. Additionally, emissions from domestic heating may also have some impact on $NH_3$ surface levels (Fig. 4f), since in the presence of non-volatile potassium cation such as from biomass burning processes, the formation of $KNO_3$ is promoted and thus the gas/particle partitioning of $NH_3$ is reduced ($12 \pm 2$ %).

## 3.3 Atmospheric water and acidity

Particle LWC and aerosol pH are important parameters of the aerosol phase; however, limited in situ observations of aerosol pH exist due to the non-conserved nature of $H^+$ and the dissociation of inorganic and organic electrolytes in the particulate phase. Since ion balance proxy methods generally lead to miscalculations of aerosol acidity levels, thermodynamic model calculations can be considered a reliable tool for such calculations (Hennigan et al., 2015). The acidity levels in an urban environment are primarily governed by the atmospheric LWC and the aqueous-phase proton concentration ($[H^+]$). The impacts of cloud LWC and aerosol water on atmospheric acidity significantly differ; cloud liquid water is primarily determined by meteorological conditions, while aerosol water is subject to chemical equilibrium with atmospheric water vapors. Thus, the hygroscopicity of the dissolved aerosol species plays a significant role in governing the aerosol water content under specific relative humidity levels (Pye et al., 2020). Moreover, the pH levels in clouds are found to be notably higher (up to four units; Seinfeld and Pandis, 2006) in comparison to aerosols, owing to the dilute conditions prevailing in them. On the contrary, aerosol acidity is determined not only by the equilibrium between acidic and alkaline species but also by the relative concentrations of such species that partition in the particulate phase under favorable temperature and relative humidity conditions (Pye et al., 2020).

Figure 5 illustrates the averaged pH levels of fine aerosols and the respective particulate water content on the surface for winter and summer, as calculated by EPISODE-CityChem. The calculated pH for fine aerosols over the GAA can be rather acidic,





with an average value of $4.81 \pm 0.93$ during winter (Fig. 5a) and $2.20 \pm 1.00$ during summer (Fig. 5b), varying spatially between 2.38 - 7.58 and 0.79 - 5.82, respectively. Minimum pH values are calculated to differ by roughly one unit, indicating an overall 10-fold increase in the aerosol $H^+$/LWC ratio from summer to winter periods in the model. Such a variation in the aerosol acidity can be directly attributed to the significantly different meteorological parameters between the two seasons, illustrating the importance of the increased relative humidity levels on aerosol pH values during the cold period. Nonetheless, the

persistence of strong acidity in fine aerosols can also be attributed to the thermodynamic equilibrium between $NH_4^+$ and $NH_3$ that overall promotes the $NH_x$ (= $NH_3$ + $NH_4^+$) volatilization in the model. This is in accordance with other studies where, although gaseous $NH_3$ may be abundant, the pH values of $PM_{2.5}$ remain acidic (Ding et al., 2019). Note, however, that for this study we only account for the water associated with the inorganic aerosol component; thus, the model may slightly underestimate the particulate water by 0.15-0.23 units compared to the pH predicted with total water (Guo et al., 2015). This

value is likely an upper bound on the error since organic aerosol mass fractions in the specific location were high (~60%; Guo et al., 2015; Stavroulas et al., 2019). On the other hand, the significantly higher maximum pH values during wintertime cannot be only attributed to meteorology. In accordance with previous studies, higher aerosol pH levels can be directly associated with the presence of NVCs emanating from wood-burning processes (Zhang et al., 2015; Bougiatioti et al., 2016). For completeness, the calculated surface aerosol water concentrations for winter (Fig. 5c) and summer (Fig. 5d) are also provided.

The long-range transport of SIA along with the emissions of its precursors, such as $SO_2$ and $NO_X$, tend to also decrease local cloud pH levels in the model. It should be mentioned that EPISODE-CityChem does not calculate, but instead only reads the cloud LWC and the cloud cover from meteorological files in the region, meaning that no transport of clouds exists in the model domain. During the winter season (Fig. 5e), higher cloud pH values have been generally calculated as compared to the summer season (not shown). The cloud pH is calculated to vary between 3.50 and 6.70 during the wintertime over Athens, mostly in

the suburbs where the higher cloudiness exists, owing to the orography of the regions. On the contrary, during the summer, the model calculates clouds of higher acidity (i.e., up to ~0.28 pH units). This, however, can be attributed mainly to the relative absence of cloud water during the warm period in the model, along with the increased transport of acidic compounds like sulfates (i.e., very concentrated solutions lead to strongly acidic pH values). Note, however, that the potential impact of mineral dust, particularly calcium, on cloud $H^+$ concentrations has been neglected in the above calculations. Such an assumption could

potentially lead to an overestimation of cloud acidity, especially during the dust event periods (mainly late winter and spring). A further constraint in the assessment of cloud acidity pertains to the exclusion of low-molecular-weight organic acids (e.g., formic and acetic acids), which may also result in some degree of miscalculation of the cloud acidity levels.

**4 Summary and conclusions**

For this work, the ISORROPIA II thermodynamic model has been coupled to the urban-scale EPISODE-CityChem CTM to

calculate the gas/particle partitioning of $NH_3$/$NH_4^+$ and $HNO_3$/$NO_3^-$, providing insights into complex interactions in a major Eastern Mediterranean urban agglomeration. While the model tends to generally underestimate the SIA measurements over



the Greater Area of Athens, it tracks their observed daily variation at Piraeus and Thissio monitoring stations. SIA formation appears to significantly depend on the ambient precursor gasses ($SO_2$, $NO_X$, and $NH_3$), which represent an important fraction (i.e., up to 20% during both cold and warm periods) of $PM_{2.5}$ in the model domain, especially in suburban areas. Nevertheless, long-range transport is found to be a critical factor in determining the SIA ambient concentrations, especially for $SO_4^{2-}$. Indeed, the model's underestimation of observed $SO_4^2$ concentrations rather indicates a too weak transport from the boundaries than a too low secondary formation from primary $SO_x$ sources. Respectively, the lower $NH_4^+$ concentrations compared to the observed values generally follow the variations of the simulated $SO_4^{2-}$, as also denoted by the derived sulfate/ammonium ratio, where sulfate is mainly in the form of $NH_4HSO_4$ and $(NH_4)_2SO_4$. Specifically, during the cold period, the relatively basic aerosol pH values enhance the volatilization of $NH_4^+$ in the model, and along with possibly missing $NH_3$ sources, the model tends to underestimate the measurements.

The inclusion of non-volatile cations originated from biomass burning sources, along with sea salt and dust aerosols, has nevertheless had a critical impact on the ion balance and thus the partitioning of semi-volatile compounds (i.e., nitrate and ammonium) in the model. Although this study is focused on the fine PM fraction, we note that high nitrate concentrations have been found to be well correlated with increased concentrations of super-micron dust and sea spray particles (Allen et al., 2015), indicating the importance of crustal species in the $HNO_3$/$NO_3^-$ partitioning. Still, besides the presence of sea salt and dust aerosols, the inclusion of $K^+$ from domestic wood-burning combustion sources in the aerosol thermodynamic calculations is found to affect the ion balance and thus the formation of $NO_3^-$, due to a more effective partitioning in the aerosol phase during the cold and humid conditions. $NO_3^-$ concentrations for 2019 exhibit a relatively lower value in comparison to the observations, with a correlation coefficient of 0.58 and a normalized mean bias of -61%. However, the model performs better in simulating $NO_3^-$ concentrations as compared to $SO_4^{2-}$ (i.e., r = 0.46) and $NH_4^+$ (i.e., r = 0.57), suggesting a satisfactory representation of the $HNO_3$/$NO_3^-$ partitioning. On the other hand, the volatilization of $NH_4^+$ due to a decrease in aerosol pH (especially during the warm period) tends to shift $NH_4^+$ to the gas phase, further indicating missing $NH_3$ sources in the model.

It is well established that SIAs play an important role in urban air quality and can have detrimental effects on the environment and human health. Thus, reducing emissions of their precursors via measures including, inter alia, the extensive use of sustainable energy, mobility, and agriculture is crucial for mitigating such effects in the long term. The continuation of such policies on the European level appears critical, especially in view of new air quality management legislation in the EU that will set very ambitious PM standards for the protection of public health, which will be difficult to attain unless sweeping emission cutbacks are implemented. Moreover, despite the fact that the reduction in primary pollutants like $SO_2$ and $NO_X$ can also reduce the acidity of cloud water and precipitation (Watmough et al., 2016), fine aerosols may still remain highly acidic in a future atmosphere due to the respective $NH_3$ volatilization in the particulate phase (Lawal et al., 2018; Baker et al., 2021). On the other hand, it is shown here that, accounting for the $K^+$ sources from domestic wood burning in the model, significantly higher aerosol pH values can be locally supported during the winter compared to the warm period, where $K^+$ is mostly associated with Saharan dust outbreaks in the GAA (Theodosi et al., 2018). Nevertheless, such a discrepancy in the response of the fine aerosol pH to emission controls also has noteworthy implications for the toxicity of atmospheric particles, especially





during the cold period where domestic wood burning may consist of an additional source of harmful air pollution. Such complex chemical interactions have not yet, however, been comprehensively acknowledged in urban-scale studies (Zhang et al., 2022).

In this study, the high spatiotemporal resolution of EPISODE-CityChem (up to 100 x 100 m², 1 h) with the improved representation of the inorganic aerosols delivers a beyond-the-state-of-the-art characterization of the fine particulate load over the GAA. Focusing further on air-quality studies in polluted urban domains, several initiatives are underway aimed at improving EPISODE-CityChem's modeling of aerosols and the respective chemical processes. These include coupling with aerosol microphysical schemes that will account for new particle formation and aerosol aging through coalescence and condensation (e.g., Karl et al., 2023). Such developments aim to improve the model's ability to represent the composition and size distribution of fine PM. A better characterization of aerosol chemical speciation, like the consideration of organic aerosols, is expected to also improve the aerosol hygroscopicity calculations in the model, especially within the boundary layer where the contribution of water-soluble organics to total aerosol mass is important (i.e., Bougiatioti et al., 2016). The KPP software for solving stiff numerical systems has already been coupled to the model to integrate the gas phase photochemistry schemes, and based on this development, more explicit chemical schemes are planned to be available in the model. Furthermore, gas-phase organic species can also be oxidized in the interstitial cloud space and form water-soluble compounds like aldehydes, which rapidly partition into droplets. In the presence of oxidants (e.g., OH and $NO_3$ radicals), the dissolved organics can also undergo chemical conversions and form low-volatile organics that remain partially in the aerosol phase. A state-of-the-art multiphase chemical scheme (e.g., Myriokefalitakis et al., 2022) is therefore planned to be coupled to the model to account for such an aerosol formation process in an urban area. Additionally, parameterizations of marine dimethylsulfide (DMS) and organic aerosol primary sources in the model are currently under consideration.

The proposed model framework, coupled with emission reduction scenarios, may enable a thorough investigation of the regulations and interventions that can efficiently curtail air pollution in the GAA and, consequently, its health impacts, providing a valuable decision support tool to the air quality management authorities in view of the realignment with the new air quality standards that will be inevitable in the near future, given the present state of air quality in Greece. Anticipated advancements in model parallelization and code architecture are expected to facilitate computationally efficient urban simulations and the creation of state-of-the-art forecasting systems capable of generating high-resolution air quality predictions for multiple urban regions in Europe up to street level. Overall, providing such an urban-scale modeling tool allows for a more accurate articulation of strategies for air quality management, including the optimization of monitoring protocols as well as the mitigation of emissions from various activity sectors and adverse effects.



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



## Tables and Figures

**Table 1. Reaction rates and constants used in EPISODE-CityChem for this study.**

| Reaction | Rate | Reference |
|---|---|---|
| | *Gas-phase chemistry rates (cm³ mol⁻¹ s⁻¹)* | |
| $SO_2(g) + OH(g) \rightarrow H_2SO_4(g) + HO_2(g)$ | $k_0 = 2.8 \times 10^{-31}(T/300)^{-2.6} \times [N_2]$ <br> $k_\infty = 2.0 \times 10^{-12}$ <br> $F_c = 0.75$ | 1 |
| | *Aqueous-phase chemistry rates (l mol⁻¹ s⁻¹)* | |
| $HSO_3^- + H_2O_2 (+ H^+) \rightarrow H_2SO_4(aq) + O_2(aq)$ | $7.5 \times 10^7 \; e^{-4430(1/T - 1/298)} \times [H^+]/(1 + 13.0 \times [H^+])$ | 2 |
| $SO_2(aq) + O_3(aq) (+ H_2O) \rightarrow S(VI)$ | $2.4 \times 10^4$ | |
| $HSO_3^- + O_3 \rightarrow HSO_4^- + O_2(aq)$ | $3.7 \times 10^5 \; e^{-5530(1/T - 1/298)}$ | 2 |
| $SO_3^{2-} + O_3 \rightarrow SO_4^{2-} + O_2(aq)$ | $1.5 \times 10^9 \; e^{-5280(1/T - 1/298)}$ | 2 |
| | *Dissociation constants (mol l⁻¹)* | |
| $SO_2(aq) \leftrightarrow SO_3^- + H^+$ | $1.3 \times 10^{-2} \; e^{1960(1/T - 1/298)}$ | 2 |
| $SO_3^- \leftrightarrow SO_3^{2-} + H^+$ | $6.6 \times 10^{-8} \; e^{1500(1/T - 1/298)}$ | 2 |
| $NH_3(aq) \leftrightarrow NH_4^+ + HO^-$ | $1.7 \times 10^{-5} \; e^{-450(1/T - 1/298)}$ | 2 |
| $CO_2(aq) \leftrightarrow HCO_3^- + H^+$ | $4.3 \times 10^{-7} \; e^{-1000(1/T - 1/298)}$ | 2 |
| $HCO_3^- \leftrightarrow CO_3^{2-} + H^+$ | $4.68 \times 10^{-11} \; e^{-1760(1/T - 1/298)}$ | 2 |
| | *Henry law constants (mol m⁻³ Pa⁻¹)* | |
| $SO_2$ | $1.3 \times 10^{-2} \; e^{2900(1/T - 1/298)}$ | 3 |
| $H_2O_2$ | $9.1 \times 10^2 \; e^{6600(1/T - 1/298)}$ | 3 |
| $O_3$ | $1.0 \times 10^{-4} \; e^{2800(1/T - 1/298)}$ | 3 |
| $CO_2$ | $3.3 \times 10^{-4} \; e^{2400(1/T - 1/298)}$ | 3 |
| $NH_3$ | $5.9 \times 10^{-1} \; e^{4200(1/T - 1/298)}$ | 3 |
| | *Heterogeneous chemistry* | |
| $SO_2(g) \; \{+ \; Dust\} \rightarrow H_2SO_4$ | $\gamma = 1.0 \times 10^{-1}$ (RH>50%) <br> $\gamma = 3.0 \times 10^{-1}$ (RH<50%) | 4 |

[1] (Atkinson et al., 2004); [2] (Seinfeld and Pandis, 2006); [3] (Sander, 2015); [4] (Liao et al., 2003)





**Figure 1. Model domain of the Greater Area of Athens, Greece. The gridded rectangles correspond to the air quality simulation domains. Sites (red dots) correspond to the air quality measurement stations during the simulation periods.**



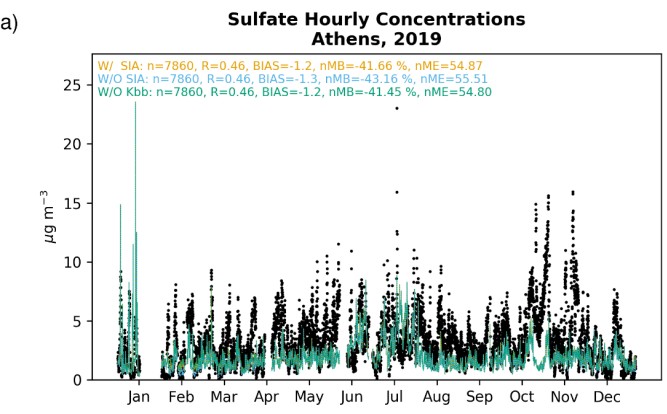

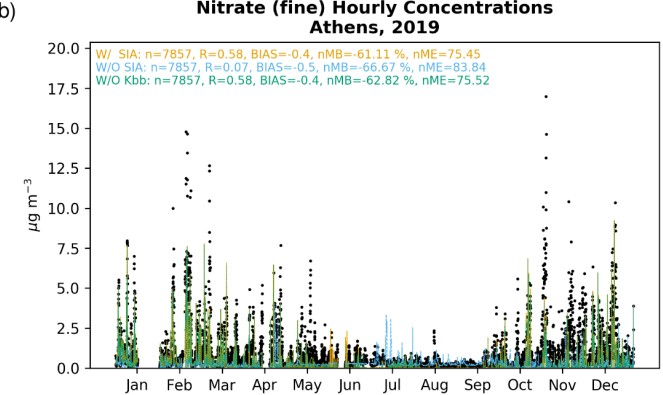

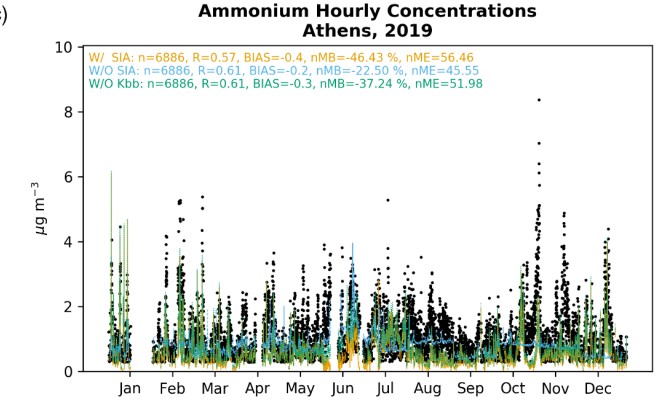

**Figure 2: Comparison (μg m⁻³) of hourly mean observations (black dots) in Piraeus (1-16 January 2019; 11 June - 9 July 2019) and in Thissio (17 January - 10 June 2019; 10 July - 31 December 2019) for a) sulfate, b) fine nitrate, and c) ammonium, to the EPISODE-CityChem simulations for base case simulation (W/ SIA; orange line), without considering SIA formation (W/O SIA; blue line) and without considering the contribution of K⁺ from domestic heating processes in thermodynamic calculations (W/O Kbb; green line). The respective correlation statistics are also provided.**








**Figure 3: Surface concentrations (μg m⁻³) over the Greater Area of Athens of sulfate (a,b), nitrate (c,d), and ammonium (e,f), as simulated by EPISODE-CityChem on a horizontal resolution of 100 x 100 m² for winter (December, January, and February; a,c,e) and summer (June, July, and August; b,d,f) of the year 2019.**







**Figure 4: Annual mean percentage (%) contribution of secondary formation processes to the surface concentrations of sulfate (a), nitrate (c), and ammonium (e), and the respective impact of including the K+ from domestic burning in thermodynamic calculations (b,d,f), as simulated by EPISODE-CityChem on a horizontal resolution of 100 x 100 m² over the Greater Area of Athens for the year 2019.**





Figure 5: Surface fine aerosol pH (a,b) and aerosol water content (c,d; µg m⁻³) for winter (December, January, February; a,c) and summer (June, July, August; b,d) 2019, along with the cloud pH at 1 km altitude for winter (e) as simulated by EPISODE-CityChem on a horizontal resolution of 1 x 1 km² over the Greater Area of Athens. The respective cloud water content (g m⁻³) is also presented (g).