# Peer review of "Analysis of secondary inorganic aerosols over the Greater Area of"

_EGUsphere, 2023_

## Author Comment (AC1)

The paper addresses an interesting topic related to the physical and chemical processes that determine the atmospheric levels of the secondary inorganic aerosols (SIA) in urban scale. The manuscript is very well written. The scientific tools used (i.e. Eulerian modeling and PM chemical composition measurements) and methodological approach are robust, while the conclusions are clear. The paper can be accepted for publication. The main minor comment is that the authors should provide maps with the chemical speciated total emission fields (precursors of SIA and SIA), so as to be easier to associate the impact of emission sources on the spatial distribution of SIA concentrations. For the same reason, the relative contribution of emission sectors on the precursors of SIA and SIA total emissions should be presented.

**Replies to RC1**

We thank the reviewer for the careful reading of the manuscript and the insightful comments. Please find below our point-by-point replies:

1. **The main minor comment is that the authors should provide maps with the chemical speciated total emission fields (precursors of SIA and SIA), so as to be easier to associate the impact of emission sources on the spatial distribution of SIA concentrations.**

**Reply:** A supplementary material document has been created for the manuscript, where a new figure (Fig. S1) has been added, depicting the spatial distribution of total emission fields of the relevant species indicated by the reviewer. A relative discussion has been added to the main text.

[Figure]

[Figure]

[Figure]

*Figure S1: Maps of total annual emission fields of: a) total NO$_x$ emissions, b) total NH$_3$ emissions, c) total SO$_2$ emissions, d) total PM$_{2.5}$ emissions, e) fine-mode sea salt emissions (SSf), and f) potassium (K$^+$) emissions from domestic burning (Kbb). The database used is CAMS regional anthropogenic emissions, which are spatially disaggregated (to 1 km$^2$) by the UrbEm approach (Ramacher et al., 2021); sea salt emissions are calculated online based on Vignati et al. (2010) parameterizations along with available updates (see text); K$^+$ emissions are derived based on non-sea salt K$^+$/PM$_{2.5}$ concentration ratios are*

*derived from filter-based PM$_{2.5}$ measurements in the center of Athens by Paraskevopoulou et al. (2014) (see text).*

**2. For the same reason, the relative contribution of emission sectors on the precursors of SIA and SIA total emissions should be presented.**

**Reply:** A new figure has been added to the supplement, depicting the contribution of emission sectors to the yearly values of SIA and their precursors as a total value for the domain of interest. A relative discussion has been added to the main text.

[Figure]

*Figure S2: Spatio-temporal totals of SIA-related tracer emissions for SO$_2$, NH$_3$, PM$_{2.5}$, Sea-Salt (expressed in NaCl), Potassium (K$^+$), SO$_4^{2-}$, and NOx (in tonnes per year per simulation domain), and the contribution of each source sector (GNFR category A: power plants; B: industrial sources; C: other stationary combustion; D: fugitives; E: solvents; F: road transport; G: shipping; H: aviation; I: off-road; J: waste; K: agriculture)*

**References**

Paraskevopoulou, D., Liakakou, E., Gerasopoulos, E., Theodosi, C., and Mihalopoulos, N.: Long-term characterization of organic and elemental carbon in the PM2.5 fraction: the case of Athens, Greece, Atmos. Chem. Phys., 14, 13313–13325, https://doi.org/10.5194/acp-14-13313-2014, 2014.

Ramacher, M. O. P., Kakouri, A., Speyer, O., Feldner, J., Karl, M., Timmermans, R., Denier van der Gon, H., Kuenen, J., Gerasopoulos, E., and Athanasopoulou, E.: The UrbEm Hybrid Method to Derive High-Resolution Emissions for City-Scale Air Quality Modeling, Atmosphere, 12, 1404, https://doi.org/10.3390/atmos12111404, 2021.

Vignati, E., Facchini, M. C., Rinaldi, M., Scannell, C., Ceburnis, D., Sciare, J., Kanakidou, M., Myriokefalitakis, S., Dentener, F., and ODowd, C. D.: Global scale emission and distribution of sea-spray aerosol: Sea-salt and organic enrichment, Atmospheric Environment, 44, 670–677, https://doi.org/10.1016/j.atmosenv.2009.11.013, 2010.

---

## Author Comment (AC2)

The manuscript describes the coupling of the EPISODE-CityChem cityscale CTM with the ISORROPIA thermodynamic model to account for SIA and acidity levels over the Greater Athens Area for a year of simulations in 2019. The manuscript is easy to follow and well-structured. However, it lacks description and discussion of two major factors that influence the interpretation of the results, being the emissions and meteorology, which I list in my below comments. The manuscript can be published in ACP when these are addressed.

**Replies to RC2**

We thank the reviewer for the careful reading of the manuscript and the insightful comments. Please find below our point-by-point replies:

**1. Line 199: The improved spatial distribution of the CAMS emissions should be shortly described.**
**Reply:** The first paragraph of Sect. 2.1.3 is now changed as below, having taken into account this comment and comments 3 to 6:
"*Anthropogenic emissions were originally provided by the Copernicus Atmosphere Monitoring Service (CAMS). CAMS provides the annual emission rates of CO, $NH_3$, NMVOCs, $NO_X$, $PM_{10}$, $PM_{2.5}$, and $SO_2$ from the road, air, rail transport, maritime (local and international shipping), mobile machinery, fuel production, industrial (paper, cement, minerals, metals, etc.), stationary combustion, agriculture, waste, solvent use, and public power sectors. As evident, forest fires or biogenic sources are not included in this dataset. CAMS adopts the GNFR sector classification and provides gridded data at 0.1 x 0.05 degrees. This dataset is then spatially refined over the domain of interest with the spatial disaggregation approach and tool documented in Ramacher et al. (2021). In particular, the regional anthropogenic dataset is combined with GNFR-dependent high-resolution spatial proxies (based on data from the European Pollutant and Transfer Register; E-PRTR, Global Human Settlement Population; GHS-POP, Corine Land Cover; CLC, and Open Street Map; OSM) towards the spatial disaggregation in 1x1 $km^2$ : i) area sources emitted 80 % in the model's layer 1 and 20 % in layer 2 for domestic heating, combustion in manufacturing industry, agriculture and farming, and other mobile sources and machinery, except for shipping, where emissions are equally distributed in the first 4 layers of the model; ii) point (industrial sources emitted at the height of each stack); and iii) line emissions (road and railway emissions emitted at 0m). We note that the top heights of layers 1, 2, 3, and 4 are 17.5, 37.5, 62.5, and 87.5 m above the ground, respectively. Annual emissions are then converted into hourly emission rates using monthly, weekly, and daily distributions based on average European factors, except for the yearly profiles for public power, refineries, and road emissions, where CAMS-TEMPO factors are applied (Guevara et al., 2021).*"

**2. Lines 212-213: Representation of sea-salt as purely NaCl vs. explicit composition could be a bit further explained. Why is it not represented with explicit composition throughout the model?**
**Reply:** Oceans have relatively similar sea salt chemical compositions, which barely change over time. Although the International Association for the Properties of Water and Steam (http://www.iapws.org/relguide/seawater.pdf) refers to the mean molar mass of sea salt as equal to ~31.403 g mol$^{-1}$, in most modeling studies, seasalt is simply expressed in the molar mass of NaCl (i.e., 58.44 g mol$^{-1}$); e.g., published seasalt emission datasets are usually expressed in NaCl (e.g., AEROCOM,

CMIP6, etc.). In this study, a specific sea salt composition is used to look at how the different minerals in sea salt affect the equilibrium calculations. According to Seinfeld and Pandis (2006) and their references, such a composition has a mean mass fraction of 55.0% $Cl^-$, 30.6% $Na^+$, 7.7% $SO_4^{2-}$, 3.7% $Mg^{2+}$, 1.2% $Ca^{2+}$, and 1.1% $K^+$. For any given molecular weight in the emissions, nonetheless, the sea salt concentrations are adjusted accordingly before the thermodynamic calculations are called in the model. However, the model uses two distinct tracers, one in the fine mode and one in the coarse mode, with a given molecular weight of NaCl to represent the emitted sea salt. Tracking the precise composition of sea salt aerosols is still feasible, but doing so would only add to the model's processing overhead (12 species instead of the current 2 accounted for). For this reason, it is a common practice in most chemistry transport and general circulation models to avoid an explicit composition of complex aerosols (such as dust and sea salt) due to a significant increase in the number of tracers transported in the atmosphere. Overall, we concur with the reviewer that a notice stating that sea salt is only represented as NaCl in the emissions may confuse the reader. For this reason, we accordingly revise the manuscript, focusing only on the explicit composition used in the thermodynamic calculations used for this study.

3. **How about biomass burning emissions other than domestic wood burning, i.e. forest fires? The area of interest is largely impacted by fires in certain periods. Same question for biogenic VOCs, are they taken into account?**

**Reply:** Indeed, Athens is expected to get transboundary influences from emissions both from forest fires (especially in the northern areas) and from regional biogenic activity. Such impacts are affecting the atmospheric air in the domain of interest through the boundary conditions of the simulation (CAMS regional reanalysis ensemble). Please also see the relevant revised paragraph in the reply to comment no. 1. Biogenic VOCs (and SOA) are outside the scope of this study. However, such an impact is extensively investigated in another paper by some of the co-authors (Karl et al., 2023), although biogenic VOC is expected to be less relevant for Athens due to the lower tree coverage.

4. **Are local and international shipping emissions taken into account? The area is highly influenced by maritime emissions.**

**Reply:** Yes, shipping emissions are taken into account in the model. Please see the relevant revised paragraph in the reply for comment no. 1.

5. **How about the vertical distribution of anthropogenic emissions? Are they all assumed to be all emitted at the surface?**

**Reply:** Area source emissions are distributed as 80 % in layer 1 and 20% in layer 2 for domestic heating, combustion in manufacturing industry, agriculture and farming, and other mobile sources and machinery at the surface, except for shipping, where emissions are equally distributed in the first 4 layers of the model. The point sources are emitted at the height of each stack, and line emissions (road and railway sources) are emitted at the surface. Please, see the relevant revised paragraph in the reply for comment no. 1.

6. **How about the temporal distribution of emissions into hourly variation represented in the model?**

**Reply:** This information is now provided in the main text. Please see the relevant revised paragraph in the reply for comment no. 1.

7.  **Section 2.1.5: Are the meteorological simulations evaluated against observations? In particular wind speed can be very challenging over urban areas. How the WRF model is setup in terms of physics should be documented.**

**Reply:** Yes, WRF simulations for the inner, high-resolution (1 km) domain have been evaluated against observations derived from 56 meteorological stations that belong to the NOA/Meteo network. A table is now added to the supplementary material of this manuscript (Table S1), showing the average statistics of temperature, pressure, specific humidity, and wind speed for all the stations. A discussion of the WRF model performance is now also added to the supplementary material (and below):

**Table S1: Average values of main meteorological parameters, including statistics against measurements from the NOA/Meteo meteorological stations in Athens.**

| Parameter | Observations Mean | Model Mean | Mean Bias | Mean Absolute Error | Correlation |
|---|---|---|---|---|---|
| Temperature (°C) | 18.5 | 18.7 | 0.2 | 1.5 | 97% |
| Pressure (hPa) | 1013.8 | 993.7 | -20.1 | 20.2 | 97% |
| Specific Humidity (g/kg) | $8.5 \times 10^{-3}$ | $8.2 \times 10^{-3}$ | $-3.0 \times 10^{-4}$ | $1.2 \times 10^{03}$ | 87% |
| Wind Speed (m/s) | 2.0 | 4.7 | 2.8 | 2.9 | 63% |

*WRF simulates the 2m temperatures in good agreement with observations, with a very low mean bias and a very good correlation. The atmospheric pressure is also well predicted, with a very good correlation. However, there is a significant bias in the pressure, which is caused by altitude deviations due to the complex landscape. These deviations are not well reproduced within the model, even at 1 km resolution. As a result, the pressure bias is close to zero in coastal stations, while in mountain stations the bias may be over 100 hPa. On the other hand, the simulated specific humidity is in line with the observations and shows a good correlation. The average normalized mean bias for all stations is -3%, indicating that the WRF slightly underestimates the atmosphere's water content. There are, however, large deviations between the modeled and observed wind speeds. The average observed wind speed is 1.97 m/s, while the modeled wind speed is 4.72 m/s for all stations. The average normalized mean bias is around 180%, indicating a significant discrepancy. This can also be seen in the high mean absolute error, while the correlation is moderate. One important factor contributing to this discrepancy is that the anemometers of the NOA/Meteo network are at a height of 5 m, while the WRF wind speed used for evaluation is at a height of 10 m. Since most of these stations are located in inhabited areas surrounded by buildings, it is expected that the wind speed changes significantly within the first few meters above the ground. Therefore, a good portion of the deviation is due to the way wind speed is measured by the NOA/Meteo network. To address this, the WRF wind speed was further evaluated against measurements from 3 stations of the Hellenic National Meteorological Service located in airports, far away from buildings, with anemometers at a height of 10 m. In these stations, the average normalized mean bias was found to be 47%. This indicates that, at least outside the urban area, the model has a real deviation much lower than what was shown in the comparison with the NOA/Meteo stations. However, there is still some*

*discrepancy, which is possibly caused by the complex geography of the area, local sea breezes, and the urban landscape.*

The WRF model setup in terms of physics is now documented in the main text as follows:

 "*WRF was setup with Lambert-Conformal map projection, two-way nesting, Thompson graupel scheme for microphysics, Monin-Obukhov (Janjic Eta) scheme for surface-layer parameterization, Noah Land-Surface model, and Mellor-Yamada-Janjic (Eta) TKE scheme for boundary layer.*"

**8. Section 3.1.1: Can the reasons for low temporal correlation (r=0.46) be explained? Are they attributed to the temporal variation of the anthropogenic emissions for example?**

**Reply:** The temporal profiles used to produce hourly emission rates from the yearly values are those provided along with the CAMS dataset for the whole European domain (see relevant text in point 1 above). Thus, the local temporal variations of emissions are not taken into account. This may lead to discrepancies between hourly concentrations from predictions and observations. A relevant discussion is now added to the respective section:

*"...(Fig. 2a). This correlation value is partly attributed to the temporal variation of emissions, which is largely based on Europe-wide factors provided along with the CAMS-REG yearly data.".*

**9. Overall, the temporal and spatial distribution and performance of the model in SIA and acidity is strongly linked with meteorology and further discussions are needed on how the WRF model represents the meteorological conditions (see comment for section 2.1.5).**

**Reply:** The following part is now added at the end of Sect. 3.1.2:

"*...of the observed values. Last, although WRF accurately predicts temperature and humidity fields, it tends to overestimate wind speed. This has the effect of enhanced ventilation and dispersion of $SO_4^{2-}$ (and the rest of SIA), which may partially explain the observed negative biases in the simulated pollutants under study.*"

**References**
https://doi.org/10.5194/essd-13-367-2021 , 2021.
https://doi.org/10.3390/atmos12111404 , 2021.
Guevara, M., Jorba, O., Tena, C., Denier Van Der Gon, H., Kuenen, J., Elguindi, N., Darras, S., Granier, C., and Pérez García-Pando, C.: Copernicus Atmosphere Monitoring Service TEMPOral profiles (CAMS-TEMPO): global and European emission temporal profile maps for atmospheric chemistry modelling, Earth Syst. Sci. Data, 13, 367–404, https://doi.org/10.5194/essd-13-367-2021, 2021.
Karl, M., Ramacher, M. O. P., Oppo, S., Lanzi, L., Majamäki, E., Jalkanen, J.-P., Lanzafame, G. M., Temime-Roussel, B., Le Berre, L., and D'Anna, B.: Measurement and Modeling of Ship-Related Ultrafine Particles and Secondary Organic Aerosols in a Mediterranean Port City, Toxics, 11, 771, https://doi.org/10.3390/toxics11090771, 2023.
Ramacher, M. O. P., Kakouri, A., Speyer, O., Feldner, J., Karl, M., Timmermans, R., Denier van der Gon, H., Kuenen, J., Gerasopoulos, E., and Athanasopoulou, E.: The UrbEm Hybrid Method to Derive High-Resolution Emissions for City-Scale Air Quality Modeling, Atmosphere, 12, 1404, https://doi.org/10.3390/atmos12111404, 2021.